# Asymptomatic immunodeficiency-associated vaccine-derived poliovirus infections in two UK children

Anika Singanayagam [1,2] ✉, Dimitra Klapsa[3], Shirelle Burton-Fanning[4], Julian Hand[1], Thomas Wilton[3], Laura Stephens[3], Ryan Mate[3], Benjamin Shillitoe [5,6], Cristina Celma [1], Mary Slatter[5,7], Terry Flood[5], Robin Gopal[1], Javier Martin [3] & Maria Zambon[1] ✉

Increasing detections of vaccine-derived poliovirus (VDPV) globally, including in countries previously declared polio free, is a public health emergency of international concern. Individuals with primary immunodeficiency (PID) can excrete polioviruses for prolonged periods, which could act as a source of cryptic transmission of viruses with potential to cause neurological disease. Here, we report on the detection of immunodeficiency-associated VDPVs (iVDPV) from two asymptomatic male PID children in the UK in 2019. The first child cleared poliovirus with increased doses of intravenous immunoglobulin, the second child following haematopoetic stem cell transplantation. We perform genetic and phenotypic characterisation of the infecting strains, demonstrating intra-host evolution and a neurovirulent phenotype in transgenic mice. Our findings highlight a pressing need to strengthen polio surveillance. Systematic collection of stool from asymptomatic PID patients who are at high risk for poliovirus excretion could improve the ability to detect and contain iVDPVs.

The oral polio vaccine (OPV) has had significant impact on the elimination of wild poliovirus (WPV) globally, as it effectively stimulates gut immunity to reduce viral transmission. Continued use of OPV in universal vaccination campaigns or as part of outbreak response is necessary as part of polio eradication in many parts of the world, with around two thirds of countries outside Europe still using the OPV. However, the risk to global polio eradication posed by continued use of live polio vaccines can be seen in the serial transmission of vaccine-derived polio in undervaccinated populations. This has resulted in the emergence of circulating vaccine-derived polioviruses (cVDPVs) in several countries, across several continents. Such viruses remain transmissible and, through multiple rounds of

person-to-person transmission, can become highly divergent from the parental OPV viruses and regain a neurovirulent phenotype causing paralysis. The first paralytic polio cases since 1989 and 2013 due to cVDPV type 3 and cVDPV type 2 have been recently reported in Israel and the USA, respectively[1]. A key factor allowing the maintenance of cVDPVs in the population is low levels of polio immunity through interrupted or low coverage vaccination programmes. As a result of declining vaccination rates, cVDPVs have been detected recently in countries that have been certified as polio free, including those that no longer use live polio vaccines (such as the USA, UK)[2–5], through the importation of individuals recently vaccinated with or exposed to OPV abroad.

[1]Polio Reference Service, UK Health Security Agency, Colindale, London, UK. [2]Department of Infectious Disease, Imperial College London, London, UK. [3]Division of Vaccines, National Institute for Biological Standards and Control, Medicines and Healthcare products Regulatory Agency, Potters Bar, London, UK. [4]Microbiology and Virology Services, Newcastle upon Tyne Hospitals NHS Foundation Trust, Newcastle upon Tyne, UK. [5]Paediatric Stem Cell Transplant Unit, Great North Children's Hospital, Newcastle upon Tyne, UK. [6]Sheffield Children's NHS Foundation Trust, Sheffield, UK. [7]Translational and Clinical Research Institute, Newcastle University, Newcastle upon Tyne, UK. ✉e-mail: anika.singanayagam@ukhsa.gov.uk; maria.zambon@ukhsa.gov.uk

A further source of divergent vaccine viruses are individuals with primary immunodeficiency (PID) in whom prolonged intestinal replication of vaccine viruses (acquired from inadvertent OPV administration, or indirectly from OPV shed by a contact) can result in immunodeficiency-associated VDPVs (iVDPVs)[6]. Globally, iVDPVs are rarely reported (only 149 cases formally reported between 1961 and 2019)[7], yet their burden is not fully understood, because current poliovirus surveillance systems are not well designed to identify iVDPV-infected PID patients who do not have neurological symptoms or paralysis. Consequently, literature is lacking on the clinical course and management of asymptomatic iVDPVs. With improving survival of PID patients, including in low- and middle- income countries, increasing use of haemopoietic stem cell transplants (HSCT) and new antiviral therapies for poliovirus in the pipeline, there is considerable interest in detailing the clinical course of asymptomatic iVDPV infection and raising awareness of inherent transmissibility of these viruses to help strengthen polio surveillance and inform case management. Here, we report on the unexpected detection of iVDPVs from two children in the UK in 2019, identified through the UK national enterovirus surveillance programme. We performed detailed clinical follow up as well as genetic and phenotypic characterisation of the infecting strains, including their intra-host evolution and assessment of neurovirulence.

## Results

### Detection and characterisation of iVDPV excreting cases

A national enhanced enterovirus (EV) surveillance programme has run for over 20 years in the UK, through which clinical laboratories voluntarily send unselected samples in which EV RNA has been detected to the national reference laboratory for typing, which is performed using partial genome sequencing of the VP1 region of enteroviruses. The first iVDPV detection in 2019 occurred when clinical materials from routine EV positive surveillance referrals from a child were found to contain a Sabin vaccine type 1-like poliovirus (SL-1) strain from a stool sample. Further viral characterisation was implemented, and clinical details sought. Child 1 (Fig. 1A) was male and born in the UK to consanguineous Middle Eastern parents. In the neonatal period, the child had a prolonged inpatient stay with enteroviral meningitis but recovered well. The child was vaccinated according to the UK schedule, with 3 doses of inactivated polio vaccine (IPV) at ages 2, 3, and 4 months. Between 6 and 12 months of age, they developed recurrent otitis media and pneumocystis pneumonia. This, alongside a family history of congenital immunodeficiency, prompted further investigation for PID. A diagnosis of CD40 ligand deficiency, a rare X-linked primary T-lymphocyte immunodeficiency, was made and the child commenced on intravenous immunoglobulin (IVIg) at 12 months of age. Between 24 and 32 months of age, the family returned to live in the Middle East where the child was treated for a further episode of pneumocystis pneumonia. Upon return to the UK, work up for HSCT was initiated during which a pre-HSCT stool surveillance sample was positive for enterovirus as well as adenovirus, norovirus genotype 2 and cryptosporidium on molecular testing in the local hospital laboratory and it was sent to the reference laboratory for typing, as part of routine national surveillance. There was no clinical suspicion for poliovirus. Characterisation at the reference laboratory identified it as an SL-1 strain, with a Ct value of 26.3. Poliovirus was isolated on L20B cell culture and confirmed on the WHO poliovirus intertypic differentiation (ITD) PCR, indicating infectious virus was being shed. Whole-genome sequencing (WGS) of the virus isolate identified 1.0% divergence in VP1 from the Sabin-1 strain, confirming excretion of a Sabin-1 derived iVDPV. The child had not received the OPV at any time and had no neurological symptoms. Acquisition of infection is presumed to have occurred during the period of residence in the Middle East in proximity to other family members vaccinated with OPV as part of national vaccination programmes. Upper respiratory tract (URT)

samples were negative for poliovirus. After 6 weeks of persistent faecal detection of poliovirus, IVIg dose was increased to improve suppression of viral replication, resulting in an increase in IgG trough level. Cessation of SL-1 poliovirus detection in stool 3 months after initial detection temporally correlated with increased IgG trough levels >10 g/dL (Fig. 2A). Increased trough levels of IVIg were maintained until HSCT was undertaken. Continued monthly testing of stool and URT samples for poliovirus by PCR and virus culture remained negative. Seven months later (age 42 months), the child underwent successful haploidentical HSCT (paternal CD3+ TCRαβ/CD19+ depleted peripheral blood stem cells (PBSC)). The post-transplant period was complicated by cytomegalovirus and adenovirus viraemia, both of which were successfully treated with antiviral therapy. Monthly samples have been negative for poliovirus for >12 months post HSCT and the child is alive and well.

Child 2 (Fig. 1B) was male and born in the Middle East to consanguineous Middle Eastern parents. Heightened awareness within the same clinical team caring for Child 1 for the potential for asymptomatic iVDPV excretion in PID patients from countries that use OPV prompted targeted screening of stool samples from Child 2, later in 2019. In the birth country, Child 2 received immunisations as per the routine childhood schedule, comprising 3 doses of IPV and one dose of OPV before 6 months of age. From age 5 months, the child experienced recurrent episodes of pneumonia, chronic diarrhoea, conjunctivitis and oral candida infections, which led to investigation for PID. Aged 16 months, they were diagnosed with a primary combined (B and T cell) immunodeficiency secondary to MHC Class II deficiency and commenced on IVIg. The child moved to the UK aged 24 months and work up for HSCT was commenced during which a stool surveillance sample tested in the local hospital laboratory was positive for EV RNA as well as Salmonella species, type 2 norovirus and coxsackie A4. As an EV RNA positive sample from an OPV-using country, the clinical team requested the stool sample be sent to the reference laboratory for EV typing. The enterovirus was identified as Sabin vaccine type 3 poliovirus-like (SL-3) virus with Ct value of 17.4 and poliovirus was successfully isolated on L20B cell culture. WGS identified 2.3% VP1 divergence from the Sabin-3 strain, confirming excretion of a Sabin-3 derived iVDPV. An URT sample taken at the same time was also positive for cultivable SL-3 poliovirus, with Ct value 30.6 indicating URT shedding, at a lower viral load than the stool. The patient had no neurological symptoms and cerebrospinal fluid examination was negative for EV RNA. Aiming to facilitate suppression of viral replication and clearance of infection, IVIg dose was increased. Despite the increase in IgG trough levels to above 10 g/dL, PV excretion continued (Fig. 2B). At age 30 months, the child underwent haploidentical HSCT (paternal CD3+ TCRαβ/CD19+ depleted PBSC). During the post-transplant period, the child required total parental nutrition to maintain nutritional status due to longstanding chronic diarrhoea. Adenovirus was detected in stool but did not require antiviral treatment. As the child's immune system reconstituted, stool pathogens were cleared. SL-3 poliovirus was cleared from the URT after day 22 post-HSCT, and from the stool after day 68. Salmonella was cleared on day 81, norovirus on day 116 and adenovirus on day 165. The child continued to excrete coxsackie A4 (CV-A4) for 4 months post HSCT. Further post-HSCT complications included a Mycobacterium bovis lymphadenitis on day 130 (treated with anti-tuberculous therapy) and an episode of intussusception on day 160 which resolved with conservative management. Monthly stool and URT samples for poliovirus culture have been negative for >12 months post-HSCT.

### Infection control and public health management

In both cases, excretion of infectious poliovirus posed a potential transmission risk, which was mitigated by enhanced infection control around the child and family. Strict isolation and infection control

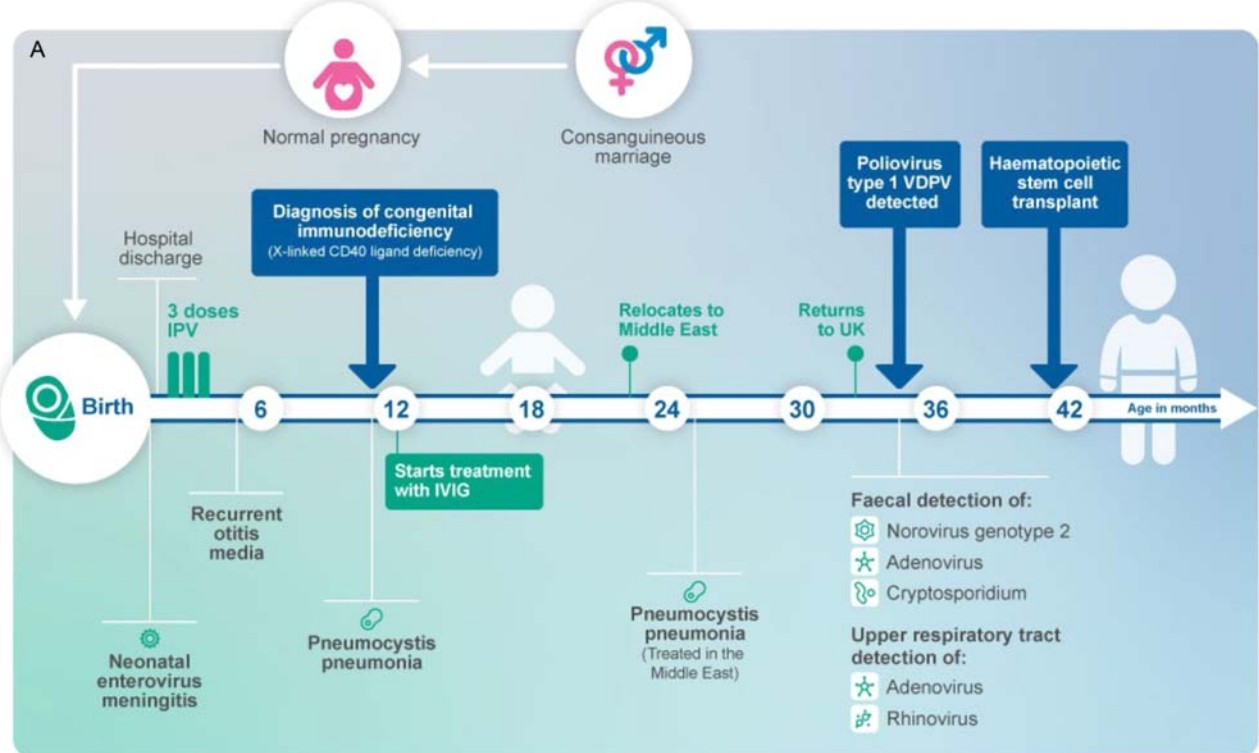

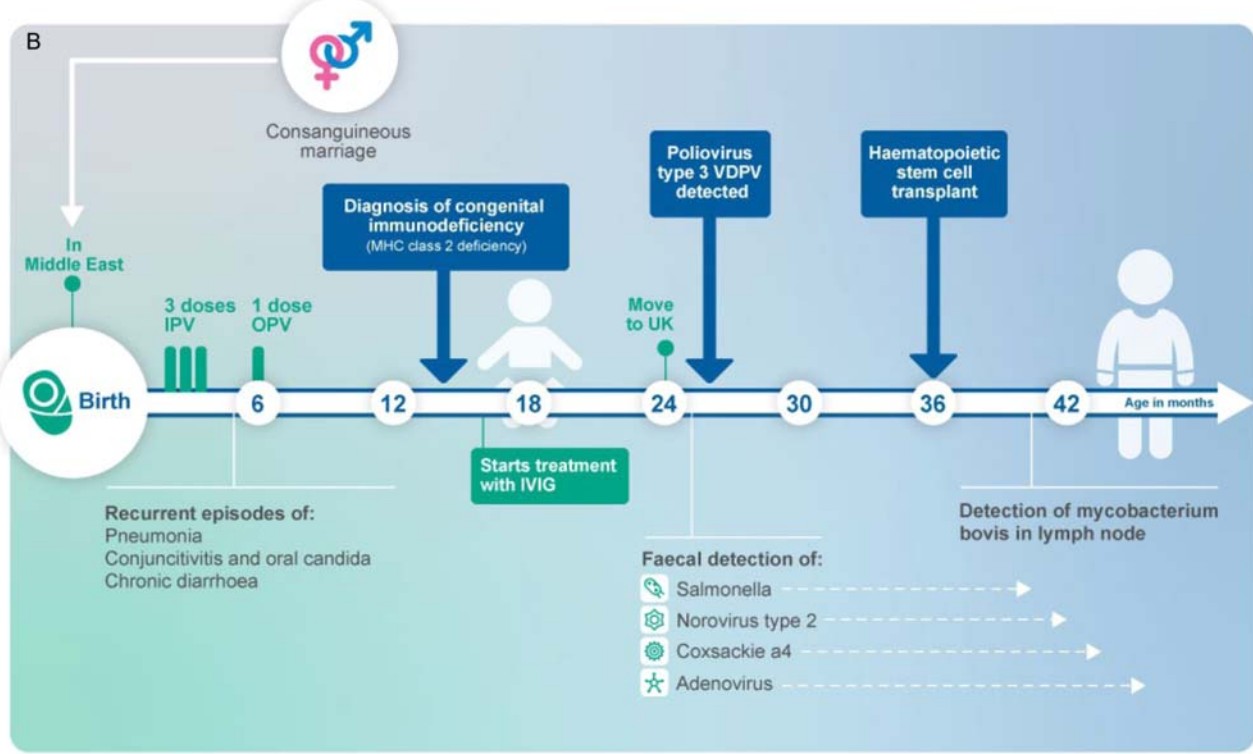

**Fig. 1 | Schematic timeline of key events and microbial detections, from birth to the post-transplant period.** Key clinical and microbiological events are shown for **A** Child 1 and **B** Child 2. *IPV* inactivated polio vaccine, *OPV* oral polio vaccine, *IVIG* intravenous immunoglobulin, *VDPV* vaccine-derived poliovirus.

protocols were initiated to reduce the risk of person-to-person transmission in the hospital and home setting. This included advice on personal hygiene in the home and adherence to enteric and droplet infection control precautions in the hospital setting. Sequential sampling from stool and URT from both cases was performed every

4 weeks until virus clearance, demonstrated by multiple consecutive negative samples (minimum two negatives, four weeks apart). In particular, Child 2 had demonstrable URT shedding of infectious virus and droplet transmission was therefore a risk. Vaccination status of staff caring for the cases on the hospital units were reviewed

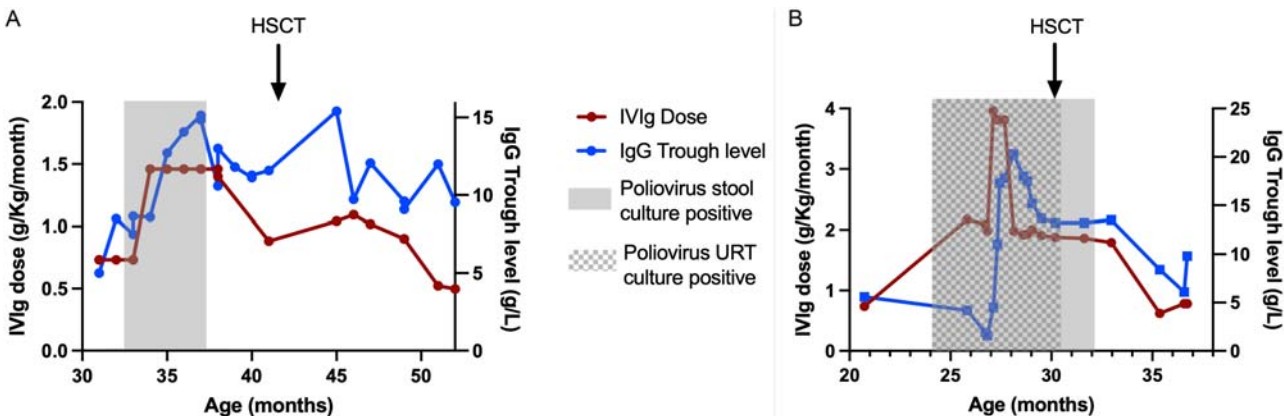

**Fig. 2 | Correlation between intravenous immunoglobulin, haematopoetic stem cell transplant and shedding of infectious poliovirus.** The relationship over time between Ig trough level (blue line), IVIg dose (red line), HSCT (black arrow), poliovirus isolation in stool (grey shading) and URT (grey and white hashed shading) is presented for **A** Child 1 and **B** Child 2. Source data are provided as a Source Data file. HSCT hematopoietic stem cell transplant, URT upper respiratory tract, Ig immunoglobulin.

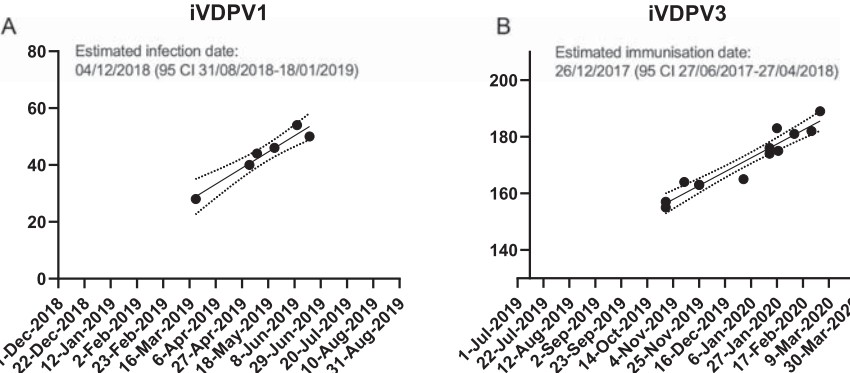

**Fig. 3 | Estimation of the date of OPV exposure from sequence divergence.** The figure represents the number of mutations from Sabin 1 and 3 vaccine strains (GenBank IDs AY184219 and AY184221). Data were adjusted to a linear function for the accumulation of nucleotide substitutions ($Y = 0.2708*X - 11764$, $R^2 = 93\%$ for iVDPV1 and $Y = 0.2328*X - 10032$, $R^2 = 92\%$ for iVDPV3). The date of initial Sabin 1 (**A**) and Sabin 3 (**B**) infection that led to the iVDPV1 and iVDPV3 cases was estimated by simple linear regression extrapolating the line for the evolution rate of nucleotide substitutions backward to 0 genome substitutions. Source data are provided as a Source Data file. OPV oral polio vaccine, iVDPV immunodeficiency-associated vaccine-derived poliovirus.

and booster IPV doses administered where required. The polio vaccination status of household members was checked and a booster dose of IPV was administered. Screening samples of stool and URT were taken from household members which confirmed absence of poliovirus shedding.

## Nucleotide sequence analysis

Poliovirus excretion was detected for 3 and 4 months since start of sampling for patients excreting PV1 (Child 1) and PV3 (Child 2), respectively. Whole-genome sequences of 6 PV1 (Supplementary Fig. 1A) and 13 PV3 (Supplementary Fig. 1B) isolates were determined by NGS. The child excreting PV3 was also observed to excrete CV-A4 for 7 months from the start of sampling. VP1 sequences from 15 CV-A4 isolates were obtained by the Sanger method and the whole-genome sequence of one CV-A4 isolate was determined by NGS (Supplementary Fig. 2).

PV1 and PV3 isolates were found to be related to Sabin 1 and 3 vaccine strains with VP1 sequence divergence ranging from 1.0–1.6% and 2.1–3.2%, respectively. Consequently, all isolates were classified as iVDPV1 or iVDPV3, i.e. having ≥1% VP1 sequence difference from the vaccine strain. Mutations showing reversion at the main attenuation site in domain V of the 5′-end non-coding region U525C and U472C for type 1 and 3, respectively, were observed in all isolates.

iVDPV3 isolates also had a change at amino acid VP3-F91S known to increase capsid stability and reduce virus attenuation. Additional mutations found, including some at known antigenic sites, are shown in Supplementary Figs. 3 and 4 for iVDPV1 and iVDPV3 isolates, respectively. We estimated the time of OPV dose initiation from the sequence divergence from Sabin 1 or 3 vaccine strains shown by the iVDPV isolates (Fig. 3) and extrapolated the regression line for the evolution rate of nucleotide substitutions back to 0 in the vaccine virus genome. This date was estimated to be 04/12/2018 (95% CI 31/08/2018-18/01/2019) for the iVDPV1 patient and 26/12/2017 (95% CI 27/06/2017-27/04/2018) for the iVDPV3 patient, consistent with exposure during time spent in a country using OPV. iVDPV3 isolates from respiratory samples had divergent sequences from stool isolates suggesting independent virus replication events (Supplementary Fig. 1). Whole-genome sequence analysis revealed a double recombinant structure Sabin 3/Sabin 1/Sabin 3 shown by all iVDPV3 isolates (Supplementary Fig. 5). Nucleotide sequence analysis of CV-A4 isolates excreted by Child 2 (Supplementary Fig. 2) also showed some degree of evolution over time.

## Neurovirulence in transgenic mice

To assess the attenuation phenotype of iVDPV1 and 3 isolates, transgenic mice expressing the poliovirus receptor were inoculated

A

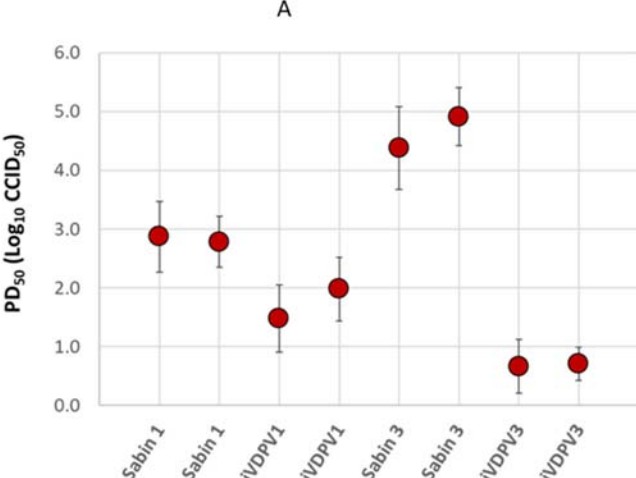

B

| Virus strain | PD$_{50}$ (95% CI) |
|---|---|
| Sabin 1 | 2.87 [2.27-3.47] |
| Sabin 1 | 2.78 [2.35-3.21] |
| iVDPV1 | 1.48 [0.91-2.05] |
| iVDPV1 | 1.98 [1.44-2.52] |
| Sabin 3 | 4.37 [3.67-5.07] |
| Sabin 3 | 4.91 [4.42-5.40] |
| iVDPV3 | 0.67 [0.21-1.12] |
| iVDPV3 | 0.71 [0.43-0.99] |

**Fig. 4 | Neurovirulence of iVDPV poliovirus strains in transgenic mice.**
**A** Transgenic mice expressing the poliovirus receptor (minimum $n = 8$ biologically independent mice per dilution) were inoculated intraspinally with stocks of representative iVDPV1, iVDPV3, Sabin 1 and Sabin 3 isolates, in duplicate. **A**, **B** The

PD$_{50}$ was calculated using the Spearman–Karber method. Data are presented as PD$_{50}$ values ± 95% confidence intervals. Source data are provided as a Source Data file. CCID$_{50}$ 50% cell culture infectious dose, PD$_{50}$ 50% paralytic dose.

intraspinally with stocks of representative iVDPV1 and 3 isolates and Sabin 1 and 3 vaccine strains in duplicate tests and observed for signs of paralysis for up to 14 days. The dose required to paralyse 50% of the mice (PD$_{50}$) was calculated using the Spearman–Karber method. As shown in Fig. 4A, B, the PD$_{50}$ for Sabin 1 and 3 in this model was 2.83 and 4.64 log$_{10}$ cell culture infectious dose (CCID$_{50}$), respectively, while iVDPV1 and 3 isolates showed lower PD$_{50}$ values of 1.73 and 0.69 log$_{10}$ CCID$_{50}$, respectively, clearly indicating a more neurovirulent phenotype as expected from their reverted genotypes.

## Discussion

In addition to the infected individual's risk of paralytic disease, iVDPV excretion represents a potential reservoir of transmissible and neurovirulent poliovirus in the post-eradication era that could act as a source of poliovirus reintroduction into the community. Whilst the WHO maintains a registry of known iVDPV cases[7,8] (mostly identified through acute flaccid paralysis (AFP) surveillance), the prevalence of asymptomatic iVDPV excretors globally remains uncertain. Many countries lack PID registries or surveillance programmes in this clinically vulnerable group. In screening studies of non-paralysed PID patients, both Sabin-like PVs and iVDPVs have been detected, though detection rates have varied from country-to-country[9–14]. Health systems increasingly support the complex care of PID patients resulting in improved survival. Higher risk countries for the emergence of iVDPVs are likely to be those with greater prevalence of PID patients, for example as a result of higher rates of consanguineous marriage, and where OPV continues to be used. Given that ~150 countries across the world continue to include OPV in their childhood vaccination programmes, exposure of PID patients to vaccine virus and subsequent prolonged excretion remains a possibility. Importation of iVDPVs into countries that have switched to IPV use (such as the UK) occurs, as was the scenario in both children reported here, who were exposed to OPV (either directly or indirectly) in countries that routinely use OPV in childhood vaccination programmes.

Although onward transmission of iVDPVs has only very rarely been documented to date, such as in reports from the USA and Spain[15,16], the risk and impact of outbreaks seeded from iVDPV shedders could increase following a reduction of population immunity. The COVID-19 pandemic has caused disruptions in polio vaccination and surveillance activities globally[17] and an upsurge in cVDPVs has been observed in recent years[18]. Given the overlap in the genetic characteristics of iVDPVs and cVDPVs[19], onward transmission iVDPVs

remains of concern. iVDPV excretion could act as a source for cryptic transmission in undervaccinated populations, to which even countries certified polio free must remain vigilant. More work is needed to delineate the molecular basis of poliovirus transmissibility in order to understand the relative transmissibility of iVDPVs compared to cVDPVs and wild type PVs. Whilst we did not detect transmission from these two cases, this is very unlikely in a well immunised country.

Until the widespread deployment globally of novel live polio vaccines that are genetically more stable[20], sensitive surveillance for VDPVs is of importance globally. Existing poliovirus surveillance systems (primarily based around AFP cases) are not well designed to identify non-paralysed iVDPV-infected patients who may shed iVDPV asymptomatically for prolonged durations. Environmental (wastewater) PV surveillance is employed for monitoring of cVDPVs but does not easily allow for identification of individual excreting iVDPV cases. In the UK, wastewater sampling was only undertaken in a limited capacity in two regions in 2019 and neither of the children were living in catchment areas for this surveillance programme. The programme has since been expanded following the detection of cVDPV in London in 2022[21]. To strengthen detection of iVDPVs, stool surveillance targeted at high-risk PID patients[22], particularly migrants from countries using OPV or known to have circulating VDPVs, could be implemented, and is a key objective of the GPEI Global Polio Surveillance Action Plan 2022–2024[23]. Indeed, the two iVDPV cases we report here were identified indirectly, the first serendipitously, through the existing UK national enhanced EV surveillance scheme without clinical suspicion for poliovirus, and the second through a targeted stool testing that occurred because, following the first case, the same clinicians had heightened awareness of the high-risk nature of the case. A key issue for routine EV surveillance in the UK has been the shift toward molecular diagnosis, particularly for using CSF, rather than acquiring stool samples to diagnose EV infections. For poliovirus, stool culture is the most sensitive detection method, which must be performed at CL3, and so detections of PV via EV surveillance are dependent upon the referral of adequate stool samples to a reference laboratory where PV isolation can be attempted.

Whilst iVDPV excretion has been reported previously, very few studies have performed monitoring and sequential sampling with genotypic and phenotypic characterisation of infecting strains. In both cases reported here, viral evolution toward neurovirulence in transgenic mice compared to the parental Sabin strain was observed,

though fortunately did not manifest clinically. Nonetheless, PID patients are reported to be at ~3000 fold increased risk of VAPP[24]. Therefore, upon detection of an iVDPV case there is a need to risk assess the individual, facilitate early viral clearance and minimise the risk of neurological symptoms. Risk assessment will include the degree of immunosuppression, the nature of virus serotype (1/2/3 or recombinant), degree of divergence from parental Sabin strain, the rate of viral evolution, the viral load, and any evidence of systemic distribution of virus for example to the URT. Here, Child 2 had a higher viral load than Child 1 and was infected with a PV3/PV1/PV3 recombinant virus displaying greater divergence from the Sabin strain and shedding from the respiratory tract also. Moreover, the child was chronically co-infected with coxsackie A4 and other pathogens, which may suggest a greater level of immunosuppression than Child 1. Of note, recombination between viral genomes is a well-known phenomenon for enteroviruses, including intertypic recombinants between different OPV strains but also recombination with certain non-polio EVs, including other coxsackie A viruses[25]. Recombination is mostly detected in Sabin2- and Sabin 3-derived isolates and rarely in Sabin 1, though sequences of Sabin 1 origin can frequently be found as PV2/PV1 and PV3/PV1 recombinant strains.

Options for the clinical management of iVDPVs are derived from case report and expert opinion, and may include administering IVIg with uptitration of the dose to achieve a higher target trough level (e.g. >10–15 g/dL), use of HSCT where appropriate, or consideration of experimental antiviral therapy. In both cases reported here, all these options were considered. In Child 1, with CD40 ligand deficiency, PV clearance was temporally correlated with an increased dose of IVIg and corresponding increase in immunoglobulin trough levels. The child later proceeded to HSCT as planned for the underlying immunodeficiency, In the second case, with combined T and B cell deficiency, increased IVIg was administered, but clearance was only later achieved shortly after HSCT. This is in line with other case reports of cessation of PV shedding after HSCT[26]. In both of the cases we report here, it was felt to be important to observe for the possibility of accelerated evolution toward neurovirulence occurring during severe immunosuppression in the peri-transplant period. For PID patients who do not clear virus, novel therapeutics can be considered[27]. At present pocapavir is the only antiviral available under compassionate use, though a combination therapy of pocapavir and V-704 is in development[28].

To conclude, declining polio vaccination rates across several countries has led to an upsurge in detection of vaccine-derived polioviruses globally[1]. In the UK, the two iVDPVs detections reported here, alongside recent genetically-linked VDPV wastewater detections in the UK, Israel and the USA[2–5], together highlight the importance of robust virological surveillance, even in countries no longer using OPV, where pockets of reduced population immunity can lead to emergence and spread of VDPVs. The highly anticipated roll out and evaluation of novel live polio vaccine strains that are more genetically stable (starting with the nOPV2, with nOPV1,3 still in development) is a matter of urgency for public health.

## Methods

### Ethics and consent

Written informed consent was received from parents/caretakers and consent was obtained to publish. The processing of patient data by Public Health England (and the successor organisation UK Health Security Agency) was conducted under Regulation 3 of The Health Service (Control of Patient Information) Regulations 2002, permitting the processing of confidential patient information for communicable disease and other risks to public health. Processing and analysis of individual patient-level data by clinicians were undertaken in compliance with the Data Protection Act. Animal work was approved by NIBSC's Ethics and Human Materials Advisory Committees. NIBSC's Animal Welfare and Ethical Review Body approved the application for Procedure Project Licence Number 70/8979 which was approved by the UK Government Home Office and under which animal care and protocols shown in this paper were conducted. All animal care and protocols used at NIBSC adhere to UK regulations (Animals, scientific procedures, Act 1986 that regulates the use of animals for research in the UK) and to European Regulations (Directive 2010/63/Eu of the European Parliament on the protection of animals used for scientific purposes).

### Laboratory detection and poliovirus isolation

Enterovirus detection was performed by rRT-PCR as previously described[29] with modifications. Briefly, following nucleic acid extraction enterovirus detection was performed using primers EVF 5′GCCCCTGAATGCGGCTAAT3′, EVR 5′AAACACGGACACCCAAAGTA3′ and probe EVPr 5′FAM-TCTGYRGCGGAACCGACT-MGB 3′, with the following cycling conditions: 15 min at 50 °C, 95 °C for 2 min followed by 45 cycles at 95 °C for 15 s and 60 °C for 1 min. Genotype assignment was achieved by partial amplification of VP1 gene and subsequent sequencing[30]. Poliovirus isolation in cell culture was performed according to WHO recommendations[31], following the WHO alternative test algorithm for poliovirus isolation and characterisation[32]. Rhabdomyosarcoma (RD), sensitive for enterovirus infection, and mouse L20B cells expressing the human PV receptor, specific for PV infection were inoculated with samples. Cells were examined microscopically for cytopathic effect (CPE) for up to 6 days. Samples with positive CPE on RD cells were passaged on L20B for specific amplification of PV. All positive L20B cell cultures were selected for further analysis. RD and L20B cells were obtained from CDC, Atlanta (USA), and are used across the WHO Global Polio Laboratory Network (GPLN) for poliovirus surveillance. Cells were authenticated by post thaw cell morphology analysis, DNA barcoding of the Cytochrome Oxidase Subunit 1(CO1) Mitochondrial Region and DNA profiling using Short Tandem Repeat (STR) profiling (RD cells only). Cells are monitored for cross-contamination following GPLN Guidance Paper 4 (Cell Authentication Testing)[33] and regularly tested for sensitivity for poliovirus infection.

### Poliovirus whole-genome next generation sequencing

Viral RNA was purified from stool suspensions or infected cell culture supernatant using the High Pure viral RNA kit (Roche). WG PV RT-PCR fragments were amplified from extracted viral RNA by one-step RT-PCR using a SuperScript III One-Step RT-PCR System with Platinum Taq High Fidelity DNA Polymerase (Invitrogen) and primers PCR-F (5′-AGA GGC CCA CGT GGC GGC TAG-3′) and PCR-3′ (5′-CCG AAT TAA AGA AAA ATT TAC CCC TAC A-3′) as described before[34]. Amplification conditions were 50 °C for 30 min for RT reaction plus 94 °C for 2 min plus 42 cycles of 94 °C for 15 s, 55 °C for 30 s and 68 °C for 8 min with a final extension step of 68 °C for 5 min. Good laboratory practice was used in all molecular assays to prevent cross-contamination of samples. Positive and negative RNA extraction and PCR controls were included in every assay. Whole-genome RT-PCR products were sequenced using Illumina protocols described before[35–37]. Sequencing libraries were constructed by A/T adaptor ligation using the KAPA HyperPrep kit (Roche, Switzerland) and dual-indexed using IDT Tru-Seq DNA unique dual indexes (Illumina, USA) with five PCR cycles for library amplification. These libraries were pooled in equimolar concentrations according to manufacturer's instructions and sequenced with 250-bp paired-end reads on MiSeq v2 (500 cycles) kits (Illumina). Initial demultiplexing was performed on-board by the MiSeq Reporter software. FASTQ sequencing data were adaptor and quality trimmed by Cutadapt v2.10[35] for a minimum Phred score of Q30, minimal read length of 75 bp, and 0 ambiguous nucleotides. NGS sequencing data were processed using Geneious 10.2.3 software package (Biomatters, Auckland, New Zealand) as described before. Briefly, raw sequence

data were imported into Geneious 10.2.3 and paired-end reads combined. FASTQ files were further processed and analysed using Geneious 10.2.3 software. Filtered reads were imported into Geneious 10.2.3, paired-end reads were combined and sequence contigs were built by reference-guided assembly. Reads were mapped to references with a minimum 50-base overlap, minimum overlap identity of 95%, maximum 5% mismatches per read, allowing up to 15% gaps, and index word length of 12. Final consensus sequences were obtained by assigning the most common nucleotide sequence to each nucleotide position within each contig assembly.

## Phylogenetic analysis of iVDPV1, iVDPV3 and CV-A4 isolates

PV1 and PV3 sequences obtained in this study were aligned to serotype 1 and 3 Sabin vaccine strain reference genome sequences (AY184219 and AY184221), respectively, using the programme ClustalW (within Geneious 10.2.3). Similarly, CV-A4 sequences were aligned to CV-A4 strain used as reference for this analysis. Bootstrapped maximum likelihood neighbour-joining trees were constructed using MEGA X software.

## Evolutionary analysis by Maximum Likelihood method

The evolutionary history was inferred by using the Maximum Likelihood method and Tamura–Nei model[38]. The tree with the highest log likelihood (−10900.70) is shown. The percentage of trees in which the associated taxa clustered together is shown next to the branches. Initial tree(s) for the heuristic search were obtained automatically by applying Neighbor-Join and BioNJ algorithms to a matrix of pairwise distances estimated using the Tamura-Nei model, and then selecting the topology with superior log likelihood value. The tree is drawn to scale, with branch lengths measured in the number of substitutions per site. This analysis involved 7 nucleotide sequences. Codon positions included were 1st + 2nd + 3rd + Non-coding. There were a total of 7441 positions in the final dataset. Evolutionary analyses were conducted in MEGA X[39].

## Transgenic mouse neurovirulence test (TgmNVT)

Tg21-Bx transgenic mice expressing the human poliovirus receptor (50% male, 50% female, 6–8 weeks old) were used for these experiments. Tg21-Bx mice are the product of crossing TgPVR21 mice with BALB/c mice, followed by repeated backcrossing of offspring with BALB/c mice, interbreeding, and selection by PCR screening of tail DNA. The mice are homozygous for PVR and class II IA b genes (H2d). Housing conditions were in compliance with the Code of Practice for the Animals (Scientific Procedures) Act 1986[40]. The mice were inoculated by the intraspinal route with 5 µl of tenfold serial virus dilutions (a minimum of 8 mice per dilution) of Sabin 1, Sabin 3 or the iVDPV1 or iVDPV3 isolates, in duplicate, and monitored for clinical signs for 14 days according to the standard operating procedure available from WHO[41]. The cell culture infectious dose ($CCID_{50}$) required to paralyse 50% of the mice ($PD_{50}$) was calculated using the Spearman–Karber method[42]. While the TgmNVT does not reproduce natural infection through the oral route, it has proven to accurately measure the neurovirulence of PV isolates and hence their potential for causing paralytic disease, showing very good correlation with the results using the gold standard monkey neurovirulence test, and is recommended by the WHO[41].

## Statistical analysis

Statistical analysis and molecular clock based inference of the date of OPV administration leading to the vaccine-derived PV (VDPV) were done using GraphPad Prism version 9 software (https://www.graphstats.net/). PV sequencing data were processed and analysed using Geneious 10.2.3 and, MEGA X software as described above. A linear regression analysis based on the number of nucleotide sequence differences from the Sabin 1 or 3 vaccine strain versus date

of sample collection was used to estimate the date of administration of the dose of oral polio vaccine leading to these infections. $PD_{50}$ in mouse assays was calculated using the Spearman–Karber method in Microsoft Excel.

## Reporting summary

Further information on research design is available in the Nature Portfolio Reporting Summary linked to this article.

## Data availability

The sequencing data (raw fastq NGS files) generated in this study have been deposited in the NCBI Sequence Read Archive under project code PRJNA924856. Nucleotide consensus sequences for poliovirus isolates are available from GenBank with accession numbers OQ286202-OQ286220. Nucleotide consensus sequences for Coxsackievirus A4 isolates are available from GenBank with accession numbers OQ319970-OQ319985. Sabin vaccine strain reference genome sequences used had GenBank accession codes AY184219 and AY184221. Source data are provided with this paper.

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

## Acknowledgements

We thank all members of the clinical and health protection teams involved in the care of the patients involved and members of the UKHSA Colindale polio reference service laboratory teams involved in processing and analysing samples. We thank Cherstyn Hurley for generating the graphics used in Fig. 1. A.S. is supported by an NIHR Academic Clinical Lectureship.

## Author contributions

R.G., J.M. and M.Z. designed the study. D.K., J.H., T.W., L.S., R.M., C.C., R.G., J.M. performed or supervised laboratory work. S.B.-F., B.S., M.S., T.F., were involved in the clinical management of patients involved in the study. A.S. drafted the manuscript. A.S., J.M., M.Z. analysed and verified the data and edited the manuscript. All authors critically revised the manuscript and approved the submitted version. All listed authors agree to all manuscript contents, the author list and its order and the author contribution statements.

## Competing interests

The authors declare no competing interests.
