## [Peer review file · Nature Communications]

REVIEWER COMMENTS

Reviewer #1 (Remarks to the Author):

I congratulate the authors on this important work on two asymptomatic iVDPV infections in two children with primary immunodeficiency. This is a topic of public health significance and highlights an at-risk population. Surveillance of individuals with PID for asymptomatic poliovirus shedding, specifically of vaccine derived strains has been previously published (<https://www.cdc.gov/mmwr/volumes/71/wr/mm7136a2.htm>, <https://doi.org/10.1093/infdis/jiu065>, Mohanty MC, Madkaikar MR, Desai M, Taur P, Nalavade UP, Sharma DK, Gupta M, Dalvi A, Shabrish S, Kulkarni M, Aluri J, Deshpande JM. Poliovirus Excretion in Children with Primary Immunodeficiency Disorders, India. *Emerg Infect Dis.* 2017 Oct;23(10):1664-1670. doi: 10.3201/eid2310.170724. PMID: 28930011; PMCID: PMC562153, Pethani AS, Kazi Z, Nayyar U, et al Poliovirus excretion among children with primary immune deficiency in Pakistan: a pilot surveillance study protocol *BMJ Open* 2021;11:e045904. doi: 10.1136/bmjopen-2020-045904). Different from other published reports is the monitoring and sequential sampling from stool and URT from both cases every four weeks, as well as nucleotide sequence analysis (though this was published in the CDC MMWR report and a few others), and neurovirulence studies.

A few specific comments on the manuscript:

- 1) I think the information on where the patient and family were residing and living needs to be better framed, as there was travel to the home region of the middle east (child one during early months of childhood) and both were born in the middle east, and child 2 moved at 24 months of age (receiving one OPV vaccine). The implications need to be further expanded here as this is not such a novel case in an endemic region of poliovirus, especially in the child who received one OPV dose.
- 2) To further elucidate the potential public health significance it is important to identify whether wastewater samples were positive during this time for iVDPV strain in the regions where the children were living. This data should be presented, or an explanation that they do not exist.
- 3) The significance of the mouse model and neurovirulence needs to also be expanded upon. Discussion on the mouse model as a role for neuroinvasive polio (importantly the virus was inoculated intraspinally) and limitations in interpretations is necessary.
- 4) Expansion on discussion of treatment modalities and choices in these cases is also important in terms of viral clearance.

Overall, the paper is well written and adds to the existing literature, though should have caveats with respect to regional public health significance given what is described as travel to and from endemic region for polio, and also lack of current data presented here on wastewater samples.

The conclusion is well supported here.

Reviewer #2 (Remarks to the Author):

In this manuscript, Singanayagam et al. report on two cases of immunodeficiency-associated vaccine derived poliovirus (iVDPV). Related to cVDPV, iVDPV arise through prolonged replication in immunocompromised hosts (typically humoral or combined immunodeficiency) with associated evolution. Over the years, ~30-50 cases of iVDPV have been reported in the literature (see Kew, Annual Reviews Microbiology and Burns, Journal of Infectious Diseases), including a case of iVDPV2 that persisted for 28 years (Javier Martin et al. PLOS Pathogens). The evolutionary pathway of iVDPV and cVDPV differ and the former are generally not known to cause outbreaks. There are cases of neurovirulence developing (see Burns, NEJM).

Here, two cases of children with severe immunodeficiency are presented. The results include the protocols that led to surveillance and identification of the cases as well as the clinical course, including clearance with increased intravenous immunoglobulin dosing and HSCT. The sequences of multiple isolates are reported with reversion of key attenuating mutations previously identified for iVDPV. A molecular clock analysis suggests a time window for acquisition of Sabin like strains in countries that still use OPV. Intraspinal infection of mice is used to measure the neurovirulence of these strains, which is increased over Sabin viruses, but thankfully was not sufficient to cause disease in these patients. The phenotype is consistent with what is known about the mutations found in the viruses in the two cases.

Overall, this manuscript presents a nice study of two cases. It doesn't really break any new ground with respect to methods to identify iVDPV, clinical risk factors, potential treatments, mutations associated with virulence, or the epidemiology of iVDPV. It serves as a timely reminder of the challenges of polio in select patient populations and surveillance issues related to the polio endgame.

Additional comments:

The analysis in Figure 3 should be clarified. Did the authors infer the time of acquisition phylogenetically (based on the tree of sequences from these patients) using root to tip distances? Or did they simply

apply a regression based on existing data on the molecular clock (number mutations fixed per unit time)? Both are valid, but have different limitations.

Page 10 C40 ligand deficiency, should be “CD40 ligand deficiency”

Supp Figure 5 - this appears to be a coverage plot for sequencing of a single isolate but legend refers to it being for all isolates. Clarify.

The sequence data should be deposited in a database and accessible.

Reviewer #3 (Remarks to the Author):

In this manuscript, Singanayagam et al report 2 children with primary immunodeficiency (PID) who were shedding vaccine-derived poliovirus without any paralytic symptoms, and were cured with either IVIG replacement and/or hematopoietic stem cell transplantation. They performed virus genome sequencing to show virus mutations that accumulated, and also demonstrated neurovirulence of the patient-isolated strains in mice. Their results led the authors to suggest that potential reservoirs of virus that could threaten public health are PID patients emigrating from regions of the world where OPV is used.

While this is certainly a topic of medical relevance, the scope of the work presented herein tends to anecdotal, being centered upon only two cases wherein there was high suspicion of poliovirus carriage to prompt culture and mutational analyses. Although the authors were constrained in their study design by the nature of how public health surveillance is currently practiced, their argument would have been more persuasive if they had performed a prospective study examining carriage of virus, mutations, and potential neurovirulence in a larger cohort PID patients stratified based upon travel or country of origin (where they would have had contact with OPV).

Conceptually, it is already known that PID patients can shed viruses, often asymptotically, whether SARS-CoV-2, norovirus, etc. or in this case poliovirus. Prolonged poliovirus shedding in particular has been previously reported for CD40 ligand deficiency (Triki et al, J Clin Microbiol , 2003) and Class II MHC deficiency (Macklin et al, Front Immunol, 2017). Unfortunately, the authors' overall argument seems a bit speculative since 1) the patients themselves remained asymptomatic despite their isolates showing increased neurovirulence in a mouse model transgenically expressing the poliovirus receptor; and 2) there was no apparent transmission from these two patients to others to demonstrate that they were

indeed the initial source of an outbreak. Given the understandably limited scope of this work, in its current format it may be better targeted to infectious diseases specialists or clinicians.

As my expertise is not in molecular virology, I cannot comment on the poliovirus mutational analyses that were performed in this study. However, it seems to me that there is insufficient information provided in the Methods about the transgenic mouse neurovirulence test, such as animal committee approval and numbers of mice used in the experiments to determine the CCID50/PD50.

RESPONSE TO REVIEWER COMMENTS

Reviewer #1 (Remarks to the Author):

I congratulate the authors on this important work on two asymptomatic iVDPV infections in two children with primary immunodeficiency. This is a topic of public health significance and highlights an at-risk population. Surveillance of individuals with PID for asymptomatic poliovirus shedding, specifically of vaccine derived strains has been previously published (<https://www.cdc.gov/mmwr/volumes/71/wr/mm7136a2.htm>, <https://doi.org/10.1093/infdis/jiu065>, Mohanty MC, Madkaikar MR, Desai M, Taur P, Nalavade UP, Sharma DK, Gupta M, Dalvi A, Shabrish S, Kulkarni M, Aluri J, Deshpande JM. Poliovirus Excretion in Children with Primary Immunodeficiency Disorders, India. Emerg Infect Dis. 2017 Oct;23(10):1664-1670. doi: 10.3201/eid2310.170724. PMID: 28930011; PMCID: PMC562153, Pethani AS, Kazi Z, Nayyar U, et al Poliovirus excretion among children with primary immune deficiency in Pakistan: a pilot surveillance study protocol BMJ Open 2021;11:e045904. doi: 10.1136/bmjopen-2020-045904). Different from other published reports is the monitoring and sequential sampling from stool and URT from both cases every four weeks, as well as nucleotide sequence analysis (though this was published in the CDC MMWR report and a few others), and neurovirulence studies.

We thank the reviewer for their comments on the importance and public health significance of our work. We have added the three references highlighted by the reviewer to our manuscript (new refs 6, 14, 22). We have also added further text in the discussion section to emphasise the unique aspects of our work as detailed by the reviewer.

A few specific comments on the manuscript:

1) I think the information on where the patient and family were residing and living needs to be better framed, as there was travel to the home region of the middle east (child one during early months of childhood) and both were born in the middle east, and child 2 moved at 24 months of age (receiving one OPV vaccine). The implications need to be further expanded here as this is not such a novel case in an endemic region of poliovirus, especially in the child who received one OPV dose.

Information about the origins and location of the children is shown in Figure 1 and also described in the text (Child 1 was born in the UK and spent 8 months in early life in the Middle East (where the family originates). Child 2 was born in the Middle East and moved to the UK aged 24 months). As suggested by the reviewer, we have added wording to the discussion to highlight the implications, and to emphasise that both children were likely exposed to OPV (directly or indirectly) in a country which routinely uses it and imported the virus into the UK.

2) To further elucidate the potential public health significance it is important to identify whether wastewater samples were positive during this time for iVDPV strain in the regions where the children were living. This data should be presented, or an explanation that they do not exist.

Thank you for this comment. At this time there was no wastewater sampling in the regions the children were living in the UK. In 2019, wastewater sampling was only undertaken in a limited capacity in 2 regions of the UK. Neither of the children were living in catchment areas for this surveillance programme, and we are unable to comment on wastewater surveillance in locations in the Middle East. We have included the information in the text.

3) The significance of the mouse model and neurovirulence needs to also be expanded upon. Discussion on the mouse model as a role for neuroinvasive polio (importantly the virus was inoculated intraspinally) and limitations in interpretations is necessary.

Transgenic mice carrying the human poliovirus receptor and therefore susceptible for poliovirus infection were developed in the 1980s and subsequently validated for measuring the neurovirulence of poliovirus isolates with different mutation profiles. Following a series of international collaborative studies, a transgenic mouse neurovirulence test (TgmNVT) was designed and validated for all three poliovirus types and is now part of the WHO recommendations for the production and control of OPV. While the TgmNVT does not reproduce natural infection through the oral route, it has proven to accurately measure the neurovirulence of poliovirus isolates and hence their potential for causing paralytic disease, showing very good correlation with the results using the gold standard monkey neurovirulence test. We have included this information and a citation (ref 38) to the WHO Standard Operating Procedure.

4) Expansion on discussion of treatment modalities and choices in these cases is also important in terms of viral clearance.

Thank you for this comment. We have expanded our discussion of treatment modalities and choices in the cases.

Overall, the paper is well written and adds to the existing literature, though should have caveats with respect to regional public health significance given what is described as travel to and from endemic region for polio, and also lack of current data presented here on wastewater samples.

Thank you - to clarify, the cases did not travel to polio endemic regions. Rather, they travelled to countries where OPV is routinely used as part of the childhood vaccination schedule and likely imported vaccine derived virus into the UK (which uses only IPV). To explain this more clearly, we have added further detail on this in the discussion which, as suggested above, helps frame the public health significance of our findings.

We have now included information about wastewater sampling in the UK, which was limited at the time of these cases.

The conclusion is well supported here.

Thank you.

Reviewer #2 (Remarks to the Author):

In this manuscript, Singanayagam et al. report on two cases of immunodeficiency-associated vaccine derived poliovirus (iVDPV). Related to cVDPV, iVDPV arise through prolonged replication in immunocompromised hosts (typically humoral or combined immunodeficiency) with associated evolution. Over the years, ~30-50 cases of iVDPV have been reported in the literature (see Kew, Annual Reviews Microbiology and Burns, Journal of Infectious Diseases), including a case of iVDPV2 that persisted for 28 years (Javier Martin et al. PLOS Pathogens). The evolutionary pathway of iVDPV and cVDPV differ and the former are generally not known to cause outbreaks. There are cases of neurovirulence developing (see Burns, NEJM).

Here, two cases of children with severe immunodeficiency are presented. The results include the protocols that led to surveillance and identification of the cases as well as the clinical course, including clearance with increased intravenous immunoglobulin dosing and HSCT. The sequences of multiple isolates are reported with reversion of key attenuating mutations previously identified for iVDPV. A molecular clock analysis suggests a time window for acquisition of Sabin like strains in countries that still use OPV. Intraspinal infection of mice is used to measure the neurovirulence of

these strains, which is increased over Sabin viruses, but thankfully was not sufficient to cause disease in these patients. The phenotype is consistent with what is known about the mutations found in the viruses in the two cases.

Overall, this manuscript presents a nice study of two cases. It doesn't really break any new ground with respect to methods to identify iVDPV, clinical risk factors, potential treatments, mutations associated with virulence, or the epidemiology of iVDPV. It serves as a timely reminder of the challenges of polio in select patient populations and surveillance issues related to the polio endgame.

We thank the reviewer for their comments, and whilst we agree that many of the observations we have made about these cases have been recorded previously in a fragmented way, the consolidation of all of these findings in two clinical cases in the UK, a country which has been considered polio free for over 40 years, emphasises the necessity for vigilance over the use of oral polio vaccines and the continued major effort that is needed for polio eradication.

Additional comments:

The analysis in Figure 3 should be clarified. Did the authors infer the time of acquisition phylogenetically (based on the tree of sequences from these patients) using root to tip distances? Or did they simply apply a regression based on existing data on the molecular clock (number mutations fixed per unit time)? Both are valid, but have different limitations.

A linear regression analysis based on the number of nucleotide sequence differences from the Sabin 1 or 3 vaccine strain versus date of sample collection was used to estimate the date of administration of the dose of oral polio vaccine leading to these infections. This has been mentioned in the Methods section and the details of this analysis are given in the Figure 3 legend.

The figure represents mean values of the number of capsid mutations from Sabin 2 vaccine strain (GenBank ID AY184220) for each of the 21 type 2-positive samples (with bars indicating standard error of the mean plotted) plotted by date of sample collection. Dashed lines show 95% CI bands of the best-fit line. The data were adjusted to a linear function for the accumulation of nucleotide substitutions ($y=0.07871*x - 3507$; $r^2=0.79$). The date of initial Sabin 2 infection that led to the type 2 isolates identified in the London sewage was estimated by simple linear regression extrapolating the line for the evolution rate of nucleotide substitutions backwards to 0 capsid substitutions.

Page 10 C40 ligand deficiency, should be "CD40 ligand deficiency"

Thank you for noting this typographical error, it has been amended.

Supp Figure 5 - this appears to be a coverage plot for sequencing of a single isolate but legend refers to it being for all isolates. Clarify.

We have modified the figure legend to make this clear.

The sequence data should be deposited in a database and accessible.

Sequence data has been deposited and is accessible. The following statement is now included in the manuscript:

"Raw fastq NGS files are available from NCBI's Sequence Read Archive under project code PRJNA924856. Nucleotide consensus sequences for poliovirus isolates are available from GenBank with accession numbers OQ286202- OQ286220. Nucleotide consensus sequences for Coxsackievirus A4 isolates are available from GenBank with accession numbers OQ319970-OQ319985".

Reviewer #3 (Remarks to the Author):

In this manuscript, Singanayagam et al report 2 children with primary immunodeficiency (PID) who were shedding vaccine-derived poliovirus without any paralytic symptoms, and were cured with either IVIG replacement and/or hematopoietic stem cell transplantation. They performed virus genome sequencing to show virus mutations that accumulated, and also demonstrated neurovirulence of the patient-isolated strains in mice. Their results led the authors to suggest that potential reservoirs of virus that could threaten public health are PID patients emigrating from regions of the world where OPV is used.

While this is certainly a topic of medical relevance, the scope of the work presented herein tends to be anecdotal, being centered upon only two cases wherein there was high suspicion of poliovirus carriage to prompt culture and mutational analyses.

We appreciate the reviewer's comment that the work is centred around two cases. These, however, are very rare occurrences – iVDPVs have only ever been detected in the UK a handful of times. Between 1961-2019, only 149 iVDPV cases were reported globally (Macklin et al MMWR 2020;69(28):913-7). The detection of these cases, particularly in light of increasing poliovirus detections globally and the detection of cVDPV in wastewater in the UK (Klapsa et al Lancet 2022; 10362:1531-8), highlights the re-emerging threat of polio. In this paper, we provide a full description of virological and clinical aspects, and report on interruption of excretion following IgG and stem cell transplant – these types of detailed analysis, with sequential sampling and linked clinical and virological data, are rarely performed and are valuable in informing case management and public health response. Given the paucity of this type of analysis in the literature, we believe that a detailed description and analysis is valuable to the international community.

Our first case was detected serendipitously via an unselected sample sent as part of voluntary laboratory enterovirus surveillance. There was no clinical suspicion of poliovirus carriage that prompted referral of the specimen and detection was unexpected. Awareness amongst clinicians for the potential for poliovirus acquisition in PID patients who may be exposed to OPV in one of the ~150 OPV-using countries is an important message of this paper. Enhancing polio surveillance in this clinical group is a key recommendation from the WHO as part of the polio eradication programme. We have added to our discussion section to strengthen these points.

Although the authors were constrained in their study design by the nature of how public health surveillance is currently practiced, their argument would have been more persuasive if they had performed a prospective study examining carriage of virus, mutations, and potential neurovirulence in a larger cohort PID patients stratified based upon travel or country of origin (where they would have had contact with OPV).

We agree with the reviewer that prospective stool collection from PID patients with linked clinical data should form part of poliovirus surveillance and is recommended in WHO guidance documents. A key message of the paper is to highlight the need to strengthen polio surveillance, including in countries deemed polio free, where importations of individuals shedding virus from countries using OPV remain a risk. As detailed above, iVDPV's are a very rare occurrence, and these cases were picked up unexpectedly through existing national surveillance structures. A key strength of our study (which is less achievable via larger scale prospective stool surveys) was the ability to obtain detailed clinical data and sequential samples from the cases. Furthermore, it is worthwhile noting that even via a prospective cohort study in PID patients in the UK, it is unlikely we would have picked up many more iVDPVs.

Conceptually, it is already known that PID patients can shed viruses, often asymptotically, whether SARS-CoV-2, norovirus, etc. or in this case poliovirus. Prolonged poliovirus shedding in particular has been previously reported for CD40 ligand deficiency (Triki et al, J Clin Microbiol , 2003) and Class II MHC deficiency (Macklin et al, Front Immunol, 2017).

Unfortunately, the authors' overall argument seems a bit speculative since 1) the patients themselves remained asymptomatic despite their isolates showing increased neurovirulence in a mouse model transgenically expressing the poliovirus receptor; and 2) there was no apparent transmission from these two patients to others to demonstrate that they were indeed the initial source of an outbreak.

Whilst we have not demonstrated transmission and disease caused by these viruses, this is very unlikely in a well immunised country. As both children were recognised to have a primary immune deficiency, infection control measures around the children and their families were considerably strengthened. Paralytic poliomyelitis occurs in <1% of poliovirus infected cases, even with wild-type strains known to be neurovirulent. However, this does not detract us from the importance of describing the risks posed by these isolates based on sequence analysis and animal studies. It is still critical to monitor and document poliovirus excretion by patients who shed VDPV asymptotically. Correspondingly, the finding of VDPV2 circulation via environmental surveillance in the UK (Klapsa et al., Lancet 2022; 10362:1531-8) is concerning despite the lack of identification of a paralytic case. The major concern is that virus will eventually reach a susceptible unimmunised individual and cause paralytic disease, as recently seen in New York.

Given the understandably limited scope of this work, in its current format it may be better targeted to infectious diseases specialists or clinicians. As my expertise is not in molecular virology, I cannot comment on the poliovirus mutational analyses that were performed in this study. However, it seems to me that there is insufficient information provided in the Methods about the transgenic mouse neurovirulence test, such as animal committee approval and numbers of mice used in the experiments to determine the CCID50/PD50.

We have included further detail in the Methods section on animal committee approval. In addition to stating the numbers of mice in the legend of Figure 4, we have also included this information now in the Methods.

REVIEWERS' COMMENTS

Reviewer #3 (Remarks to the Author):

The authors have persuaded me of the value of thie report despite the few cases, retrospective nature, and limited novelty, given the rarity of the cases and the public health impact.